# Insight into the Chromosome Structure of the Cultivated Tetraploid Alfalfa (*Medicago sativa* subsp. *sativa* L.) by a Combined Use of GISH and FISH Techniques

**DOI:** 10.3390/plants9040542

**Published:** 2020-04-22

**Authors:** Egizia Falistocco

**Affiliations:** Department of Agricultural, Food and Environmental Sciences, University of Perugia, Borgo XX Giugno, 06121 Perugia, Italy; egizia.falistocco@unipg.it; Tel.: +0039-075-585-6209; Fax: +0039-075-585-6224

**Keywords:** Medicago sativa, FISH, GISH, C-banding, repetitive DNA, telomeres

## Abstract

Cytogenetic research in *Medicago sativa* subsp. *sativa* L., the cultivated tetraploid alfalfa (2n = 4x = 32), has lagged behind other crops mostly due to the small size and the uniform morphology of its chromosomes. However, in the last decades, the development of molecular cytogenetic techniques based on in situ hybridization has largely contributed to overcoming these limitations. The purpose of this study was to extend our knowledge about the chromosome structure of alfalfa by using a combination of genomic in situ hybridization (GISH) and fluorescence in situ hybridization (FISH) techniques. The results of self-GISH (sGISH) suggested that a substantial part of the repetitive fraction of the genome of subsp. *sativa* is constituted by tandem repeats typical of satellite DNA. The coincidence of sGISH and C-banding patterns supported this assumption. The FISH mapping of the *Arabidopsis*-type TTTAGGG telomeric repeats demonstrated, for the first time, that the alfalfa telomeres consist of this type of sequence and revealed a massive presence of interstitial telomeric repeats (ITRs). In the light of this finding *M. sativa* appears to be a suitable material for studying the origin and function of such extra telomeric repeats. To further exploit this result, investigation will be extended to the diploid subspp. *coerulea* and *falcata* in order to explore possible connections between the distribution of ITRs, the ploidy level, and the evolutionary pathway of the taxa.

## 1. Introduction

*Medicago sativa* subsp. *sativa* L. (2n = 4x = 32), the cultivated alfalfa, is one of the most important forage crops in the world and the most intensively studied species of its genus. Despite this, cytogenetic research on it has lagged behind other crop species mostly due to the characteristics of its chromosomes, which are small and very uniform in shape and size, thus making it difficult to perform detailed analyses. However, progress achieved in molecular cytogenetic techniques based on in situ hybridization, such as fluorescence in situ hybridization (FISH) and genomic in situ hybridization (GISH), has opened new opportunities for the study of plant species with this type of chromosome [1]. For instance, the FISH procedure using simultaneously 45S and 5S rDNA probes provided the first cytogenetic markers for chromosome identification in the tetraploid subsp. *sativa,* the related diploid subspp. *coerulea* and *falcata*, and the ancestral species *Medicago glomerata* [2]. Recently, the hybridization of different repetitive DNA sequences produced additional markers for a more detailed karyotype analysis [3]. However, the total potential of in situ hybridization techniques has not been so far fully exploited on alfalfa. For example, the GISH procedure, a very effective way for examining the organization of repetitive DNA in plant genomes, has never been applied. It is known that a major component of the plant nuclear genome consists of repetitive DNA sequences that are typically subdivided into two major categories—one including DNA repeats that are scattered all throughout the genome and the other comprising copies arranged in tandem arrays of the basic motif. Among these, the satellite DNA (satDNA) represents the most abundant fraction. It consists of highly repeated DNA sequences located at the heterochromatic regions of the chromosomes, mainly in centromeric and subtelomeric regions, but also at intercalary sites [4,5].

Both tandem and dispersed repeats are the sequences mainly involved in the hybridization reaction; therefore, the GISH method effectively examines the organization of these sequences [6]. The self-GISH (sGISH), a variant of GISH consisting in the hybridization of the genomic DNA of a species to its own chromosomes, is a feasible and rapid method for investigating the chromosome distribution of the repetitive DNA of a species by avoiding more complex and expensive procedures such as the isolation and physical mapping of different repetitive DNA families or the sequencing of whole genomes. The patterns of hybridization generated by the genomic probes reproduce the position of the chromosome regions where repetitive DNA is concentrated and offer indications of the prevalence of dispersed or tandem repeats in the genomes [6,7,8].

The GISH procedure has been applied in many fields of plant cytogenetics including chromosome structure, karyotyping, genomic affinity, and phylogenetic relationships [9,10,11,12]. Considering all this, the present study was undertaken to gain further knowledge about the chromosome structure of *M. sativa* subsp. *Sativa*, by obtaining additional information on the repetitive fraction of its genome. As a first step, the sGISH procedure was used to visualize the overall distribution of repetitive DNA. Then, to characterize the genome in more detail, a comparison between molecular and conventional cytogenetic data, obtained by the C-banding procedure, was made. Finally, FISH was applied to verify if and where telomeric (TTTAGGG)n arrays are distributed in the alfalfa chromosomes. At the moment, there is no precise evidence of the molecular composition and chromosome localization of telomeric sequences in alfalfa. The majority of plant species thus far investigated exhibit *Arabidopsis*-type TTTAGGG telomeric repeats. However, there are several exceptions; for example, some species of *Allium* (Asparagales) and *Cestrum* (Solanaceae) have been found to lack this type of sequence [13,14]. With progressive research, it has become evident that telomeric repeats are not exclusive of the chromosome ends. A considerable number of species have been found with telomere-like repeats also positioned in internal regions of chromosomes [15]. Such internal repeats, now currently identifiable as interstitial telomeric repeats (ITRs) or interstitial telomeric sequences (ITSs), are of particular importance in studying karyotype evolution and genome plasticity in several animal and plant taxa [16,17].

## 2. Materials and Methods

### 2.1. Plant Materials

The materials used for this study consisted of seeds of *M. sativa* subsp. *sativa* belonging to the commercial variety Pomposa and to the ecotype Central Italy. The seeds were germinated at 20–22 °C in Petri dishes on filter paper moistened with distilled water. Some of the resulting seedlings were utilized for cytological preparations, while others were transplanted into Jiffy pots and transferred to the greenhouse where the plants were grown, under controlled conditions, for the production of leaves.

### 2.2. DNA Extraction and Preparation of Probes

For the preparation of the genomic probe, total genomic DNA was extracted from young leaves of subsp. *sativa* using the Qiagen mini kit (Qiagen Ltd., Manchester, UK) according to the manufacturer’s instructions. Then, DNAs were labelled with biotin-11-dUTP (Sigma Aldrich Co., Steinheim, Germany) or digoxigenin-11-dUTP (Roche, Indianapolis, IN, USA) by nick translation. The plasmid pUc18 containing the sequences (TTTAGGG)n of *Arabidopsis thaliana* was labelled with biotin-11-dUTP by PCR for identification of the telomere repeat sequences (TRSs). Clone pTa71 containing a 9-kb *Eco*RI fragment of the 18S-5.8S-25S rRNA genes and the intergenic spacers from wheat [18] was labelled with digoxigenin-11-dUTP and used to localize the sites of 45S rDNA.

### 2.3. Chromosome Preparations and C-Banding

The procedure for the preparation of mitotic chromosomes used for GISH, FISH, and C-banding was essentially the same as that described in detail in Falistocco et al. [12]. Briefly, whole seedlings with roots 1.0–1.5 cm long were immersed in ice-cold water and incubated for 24 h at 4 °C in order to accumulate metaphases. After this treatment, the seedlings were immersed in a saturated aqueous solution of α-bromonaphthalene for 2–3 h, fixed overnight in 3:1 absolute ethanol/glacial acetic acid at room temperature and then stored at −20 °C until required.

Excised roots were washed in distilled water for 10 min and treated for 1–2 min in 0.25 N HCl, rinsed again in distilled water, and transferred to enzyme buffer (10 mM citric acid/sodium citrate, pH 4.6) for 20 min. Root tips were then excised and digested in the enzyme solution (4% cellulase Onozuka R10 and 1% pectolyase (Sigma-Aldrich, Co., Steinheim, Germany) in distilled water) for 2 h at 37 °C. The cell suspension was pelleted and resuspended in enzyme buffer. After pelleting, the material was washed two times with the fixative and resuspended in the fixative. Finally, 20 to 30 μL of cell suspension were applied to a slide. The C-banding method was basically that described by Falistocco et al. [19]. The slides were immersed in a saturated solution of Ba(OH)_2_ for 5 min at 35 °C, rinsed in distilled water, incubated in 2× SSC ( 0.3 M NaCl, 0.03 M sodium citrate) at 60 °C for 15–20 min, and then stained with 2% Giemsa for 10 min.

### 2.4. Fluorescence In Situ Hybridization

sGISH was performed following the protocol described by Falistocco [12]. The slides were pretreated with 100 μg/mL of RNase A in 2× SSC (0.3 M NaCl, 0.03 M sodium citrate) for 1 h at 37 °C and washed three times in 2× SSC. After incubation with 80 units/mL of pepsin (Sigma-Aldrich) in 10 mM HCl for 15 min at 37 °C, the chromosome preparations were stabilized by immersion in freshly depolymerized 4% (*w*/*v*) paraformaldehyde in water for 10 min at room temperature, washed in 2× SSC, dehydrated in a graded ethanol series, and air-dried. The hybridization solution was similar for FISH and GISH experiments with the exception of the concentration of probes and the salmon sperm DNA that was not used for GISH. It consisted of 2 ng/μL of ribosomal and telomeric probe or 5 ng/μL of the genomic probe, 50% (*v*/*v*) formamide, 10% (*w*/*v*) dextran sulphate, 0.1% (*w*/*v*) sodium dodecyl sulfate (SDS), and 300 ng/μL sheared salmon sperm DNA. The hybridization mixture was pre-denatured for 10 min at 70 °C and chilled on ice, then it was applied to the slides and denatured together with chromosome preparations at 70 °C in a modified thermocycler for 5 min. Then, the temperature was gradually decreased to 37 °C. The hybridization was carried out overnight at 37 °C. Post hybridization washes were carried out in 20% formamide (*v*/*v*) in 2× SSC at 42 °C. The same procedure as described for GISH was used for in situ hybridization of clone pTa71 and telomeric sequences labelled with digoxigenin and biotin, respectively. Detection of digoxigenin- and biotin-labelled probes was carried out with anti-digoxigenin conjugated with FITC (Roche) and streptavidin conjugated with Cy3 (Sigma-Aldrich), respectively.

All the preparations were counterstained with 2 μg/mL of 4′,6-diamidino-2- phenylindole (DAPI) and then mounted in antifade solution Vectashield (Vector Laboratories, Petergorough, UK).

Karyograms were obtained by arranging the chromosomes into eight homology groups and disposing the groups according to decreasing length, with the exception of the satellite chromosomes which were assembled as group 8 regardless of length.

The slides were examined with a Microphot Nikon epifluorescence microscope (Tokyo, Japan). Images were recorded with a SONY digital photocamera (Tokyo, Japan) and then processed using Adobe Photoshop CS3 (San Jose, California, USA).

## 3. Results

### 3.1. sGISH and C-Banding in M. sativa subsp. sativa

The chromosome structure in subsp. *sativa* was investigated based on the distribution patterns of repetitive DNA and heterochromatin by using a combination of sGISH and Giemsa C-banding procedures. No substantial differences were observed among the examined accessions for both sGISH and C-banding, and the mitotic metaphase chromosomes of the alfalfa ecotype are displayed in Figure 1 and Figure 2 illustrating the obtained results.

### 3.2. sGISH Signal Pattern

A mitotic metaphase of alfalfa after the sGISH treatment and the relative karyogram are displayed in Figure 1. All the examined metaphases clearly showed that the fluorescence was not uniformly distributed along the chromosomes but was concentrated in distinct regions forming discrete bands (Figure 1a,b). For this reason, the hybridization pattern may be referred to as GISH banding. Two strong signals were observed in all chromosomes, one in the centromeric position and the other at the terminal part of the short arm, whereas the long arm end did not show any sign of hybridization. Additional bands were also noticed. An interstitial fluorescent band was located in the short arm of all chromosomes with the exception of the smallest ones (homology group 7). Another band was localized proximally in the long arm of the largest chromosomes of the complement (homology group 1). The interstitial bands present in the short arms were clearly identified in chromosomes in the early metaphase stage. In more contracted chromosomes, they tended to be associated with the centromeric bands especially in the chromosomes of medium or small size (Figure 1b). The nucleolar chromosomes, arranged in group 8, displayed a fluorescent pattern characterized by strong signals at the centromeric region and two other signals in the arm not carrying the nucleolar organizer region (NOR), one close to the centromere and the other in terminal position. Faint traces of fluorescence were seen in correspondence to the secondary constrictions (Figure 1b).

### 3.3. Giemsa C-Banding Pattern

A C-banded mitotic metaphase of alfalfa and the respective karyogram are shown in Figure 2. The C-banding procedure revealed extended regions of heterochromatin in the chromosome complement of alfalfa with a prominent band at the centromeric region and a further band at the end of the short arm of all chromosomes (Figure 2a). Intercalary heterochromatic bands were also observed. They were more or less clearly identified depending on the level of contraction of the chromatin fiber. For chromosomes in the early metaphase stage, such as those displayed in Figure 2, an interstitial band located in the short arm was clearly identified in all the chromosomes, except chromosomes of group 7. Another band localized in proximal position of the long arm was observed only in chromosomes of group 1 (Figure 2b). The nucleolar chromosomes exhibited an obvious band at the centromere and two bands in the arm not carrying the NOR, one in proximity of the centromere and the other at the chromosome end. Heterochromatic regions were also localized at both sites of the secondary constrictions (Figure 2b).

### 3.4. Comparison between sGISH and C-Banding Patterns

When compared with C-banding, the sGISH pattern revealed strong hybridization signals in the C-band regions and weaker signals or no signal at all in the remaining parts of the chromosomes. The only exception consisted of the portions of the chromosome arm and satellite adjacent to the secondary constrictions, which did not show considerable trace of fluorescence but exhibited consistent positive C-bands.

### 3.5. FISH Mapping of Telomeric (TTTAGGG) Repeats

The plant-type telomere probe efficiently hybridized to the extremities of all chromosomes, thereby indicating the presence of the TTTAGGG sequence in the alfalfa telomeres (Figure 3). All telomeres were distinctly labelled in each metaphase spread with no noticeable differences in signal intensity among chromosomes. Interestingly, in addition to the telomeres, the probe revealed numerous telomeric repeats clustered at interstitial sites of chromosomes mostly in pericentromeric regions. The number of chromosomes carrying ITRs was difficult to establish; however, by examining numerous metaphases, it could be estimated that at least 12 chromosomes have such extra telomeric sites. Similarly to the telomeric sites, the ITR sites were visualized as pairs of fluorescent spots positioned on sister chromatids. A few chromosomes of medium size exhibited a pair of fluorescent spots in both chromosome arms. Interstitial telomeric sites were found also in nucleolar chromosomes in proximity of the secondary constrictions (Figure 3). No substantial differences of hybridization signal intensity among interstitial sites were observed. Taking into consideration the intensity of the signals, it may be presumed that the ITRs visualized are organized into quite long arrays of similar length.

## 4. Discussion

To know the relative disposition of genes and repetitive sequences is of crucial importance for studying the evolution and behavior of plant genomes. The banding patterns generated by the hybridization of genomic probes in several plant species have been explained by the presence of highly repetitive DNA sequences, which are clustered at specific regions of chromosomes and appear as distinct bands after labelling [9,10,11]. Moreover, the coincidence of GISH banding and C-banding [7,20,21,22,23] demonstrated that the discrimination of the heterochromatin by sGISH is essentially based on the abundant and highly repetitive DNA present within the heterochromatic blocks. The results of this study, based on sGISH and C-banding, revealed that a substantial part of the repetitive fraction of the genome of *M. sativa* subsp. *sativa* is constituted by tandem repeated sequences typical of satDNA. The close correspondence between the sGISH and C-banding patterns, which excludes only small areas adjacent to the secondary constrictions, further supports this assumption. This discrepancy could be explained by the presence in the heterochromatin surrounding the NORs of moderately repetitive sequences, such as transposable elements, which under general experimental conditions are not involved in the hybridization reaction [4,24]. On the other hand, telomeres and NORs were also not labelled by the genomic probe, probably due to the copy number of the telomeric and rDNA repeats, which is not as high as that of satDNA.

The interesting outcome of this study concerns the localization of telomeric repeats. Using FISH, it was possible to display, for the first time, the presence and distribution of telomeric (TTTAGGG)n repeats on alfalfa chromosomes. A mention of the telomeric sequences of alfalfa has been made by Yu et al. [3]. However, the sequences that these authors used for FISH experiments produced signals exclusively at the ends of chromosomes. The FISH mapping of the telomeric repeats carried out during this study pointed out a few important aspects that are worth considering. The first is that the chromosome termini of alfalfa showed strong homology with the telomeric *Arabidopsis* sequence TTTAGGG, indicating that alfalfa shares this sequence with most of the plant species so far investigated. An extensive study carried out on telomere sequences in higher plants revealed that telomeric repeats in most angiosperm and gymnosperm plants as well as in bryophytes are represented by the heptanucleotide motif TTTAGGG, suggesting that this sequence is highly conserved and that it probably represents the basic telomere sequence of higher plant phyla [13].

The second important result is the finding of telomeric repeats in interstitial positions. By definition, telomeric sequences are located at the ends of chromosomes; however, over the years, an increasing number of plant species with ITSs have been discovered [13,16]. It is well known that telomeric DNA repeats play a key role in chromosome stability, preventing end-to-end fusions and precluding the recurrent DNA loss during replication. On the contrary, the full meaning of the interstitial telomeric repeats is still not totally clear and many hypotheses on their origin and function have been made. Commonly, ITSs are considered a valuable indicator for elucidating mechanisms of karyotype evolution and changes in chromosome number. For example, interstitial telomeric repeats (ITRs) often represent a relic of the fusion of two ancestral chromosomes resulting in chromosome number reduction [15,16,25]. However, other theories on their origin have been proposed. For example, mechanisms of double-strand break repair or transposition from telomeric region arrays to other regions of the chromosomes [26]. Most of the non-telomeric sites identified in alfalfa during this study were localized at centromeric or pericentromeric regions. Based on cytogenetic analyses in vertebrates, the ITSs having this chromosome localization constitute a specific group characterized by large arrays usually positioned within or at the margin of heterochromatin. For this reason, they are named “centromeric” or “pericentromeric” ITSs or, more properly, “heterochromatic” ITSs. The biological function of these sequences is not clearly understood. They seem to have a role in karyotype evolution, but it has been supposed that in several cases their presence in the chromosomes simply reflects the fact that these sequences are a component of the centromeric satellite DNA [17]. The results of this study suggest that most of the ITRs discovered in subsp. *sativa* belong to this category. The extra telomeric sites identified by FISH appeared to be located within the heterochromatin surrounding the centromeric regions. The intensity and size of the fluorescent signals indicate that such telomeric repeats are organized into long arrays. The localization of telomeric repeats in proximity of the nucleolar constrictions, as in alfalfa, has also been found in other organisms. That telomeric sequences and specifically ITSs might play a role in nucleolus organization is an accepted possibility [27]. The results of this study indicate that *M. sativa* subsp. *sativa* could be a suitable experimental material for use in further investigations on the origin and function of ITRs. By extending the study to the diploid subspp. *coerulea* and *falcata*, it may be possible to investigate possible correlations between the distribution of ITSs, the ploidy level, and the evolutionary pathway of the taxa.

## Figures and Tables

**Figure 1 plants-09-00542-f001:**
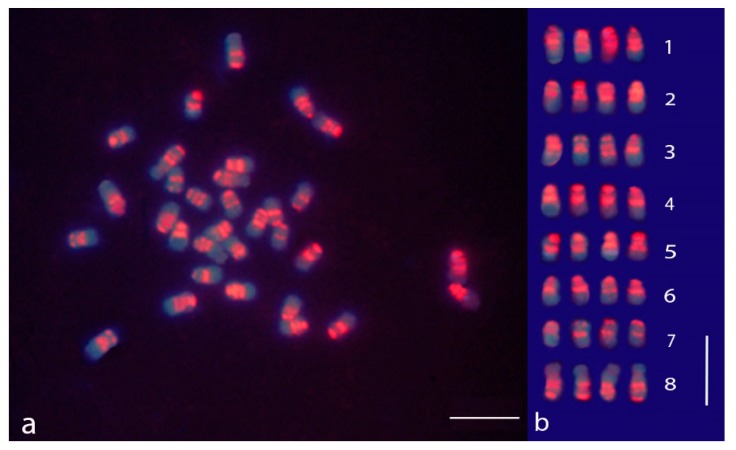
Self-genomic in situ hybridization (sGISH) banding in chromosomes of *M. sativa* subsp. *sativa* displaying the localization of highly repeated DNA sequences (satDNA). (**a**) Mitotic metaphase after the sGISH treatment; (**b**) karyogram of the sGISH-banding metaphase shown in Figure 1a; the chromosomes were arranged according to their morphological traits and the sGISH signal pattern. The bar represents 5 µm.

**Figure 2 plants-09-00542-f002:**
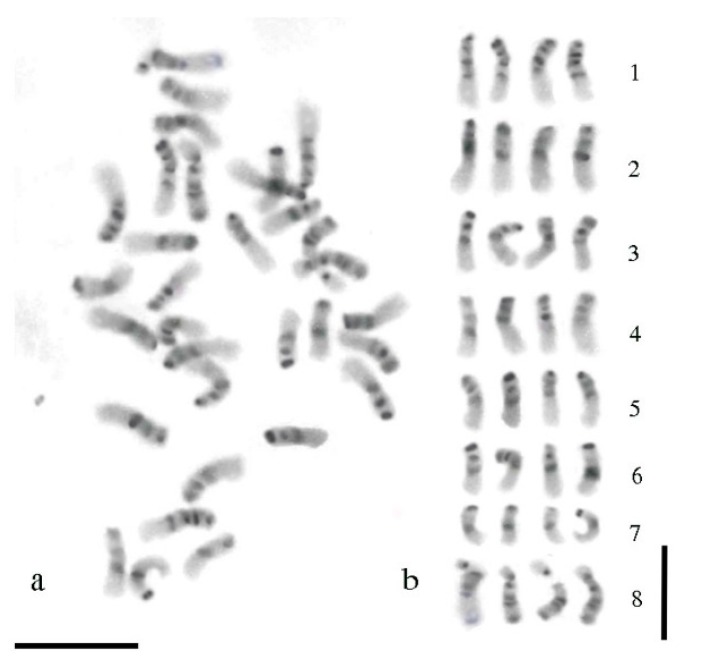
C-banding in the tetraploid alfalfa. (**a**) Mitotic metaphase chromosomes after the C-banding treatment and Giemsa staining; (**b**) karyogram according to Figure 1a; chromosomes were arranged according to their morphological traits and distribution of the heterochromatic bands. The bar represents 5 µm.

**Figure 3 plants-09-00542-f003:**
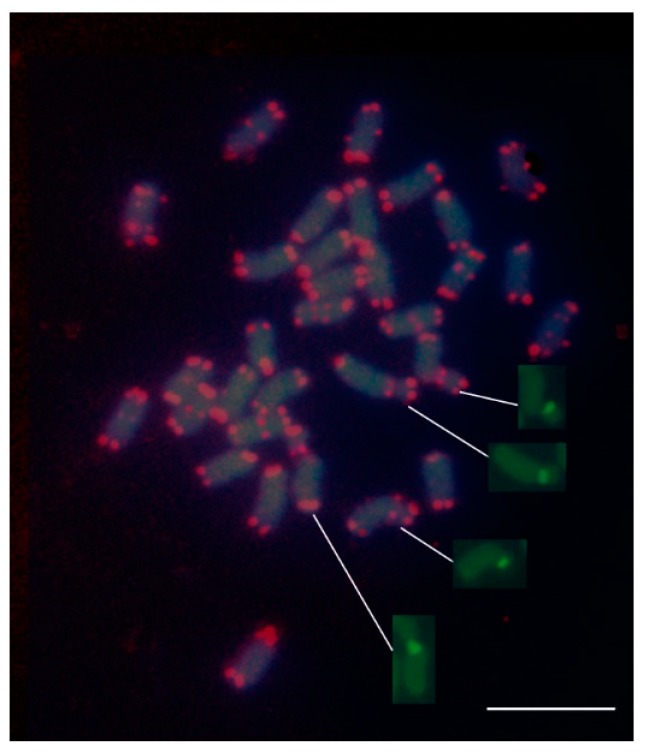
Fluorescence in situ hybridization (FISH) mapping of telomeric (TTTAGGG)n sequences and 45S rDNA on mitotic metaphase chromosomes of the tetraploid alfalfa. The extremities of all chromosomes were mapped by the telomeric probe. Note the presence of interstitial telomeric sequences (ITSs) at the centromeric regions of numerous chromosomes; ITSs were also localized in correspondence to nucleolar organizer regions (NORs) (arrows). Insets show the nucleolar chromosomes with signals of the 45S rDNA probe without being overlapped with DAPI. The bar represents 5 µm.

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
