# Peer review of "Insight into the Chromosome Structure of the Cultivated Tetraploid Alfalfa (Medicago sativa subsp. sativa L.) by a Combined Use of GISH and FISH Techniques"

_plants, 2020, doi:10.3390/plants9040542_

Round 1
Reviewer 1 Report
This is a rather small-scale and simple piece of molecular cytogenetic work on chromosomes of a tetraploid cultivar of an important legume forage crop Medicago sativa. Although its methodology is far away from being ground-breaking in plant molecular cytogenetics AD2020, this study provides some novel and interesting data regarding chromosomal distribution of repeats demonstrated by GISH and FISH with telomeric probe and fills some gaps regarding our knowledge of a chromosome and karyotype organisation in alfalfa. Distribution of (TTTAGGG)n sequence at quite numerous extratelomeric chromosome locations is nicely and convincingly demonstrated. Discussion is not very deep but sufficient. Below are some, essentially editorial suggestions and comments:
- L169/L170: ‘(…) was clearly identified in all the chromosomes, a part the chromosomes of group 7.’. I find the meaning of this part of the sentence unclear. Consider rewording, e.g. ‘except chromosomes of group 7’ instead ‘a part the chromosomes of group 7’?
- L173 and elsewhere in the MS: the word (chromosome) ‘extremity’ is not often used by cytogeneticists. Chromosome terminus, chromosome end, distal (terminal) part of a chromosome / chromosome region are more common alternatives.
- 191: consider adding (ITR) here. I know that this abbreviation was defined in the Abstract but as far as I see it was not in the main text.
- 194: Change ‘ITRs sites’ to ‘ITR sites’.
- L231: Change ‘chromosome terminals’ to ‘chromosome termini.
- Fig. 1b and 2b: consider scale bars to be oriented vertically not horizontally.
- References: year of publication is missing for the paper #2 and it is not in bold for #17.
Author Response
My response to observations of Reviewer 1 is the following:
- L 169/170 – the sentence indicated was changed as suggested: “except chromosomes of group 7”.
- L 173 – the term “extremity” was substituted by “chromosome end” , at L 146 by “end” .
- L 191 – on this point please observe that this abbreviation was defined also at the end of the INTRODUCTION. However, if necessary it can be added to the point indicated by the Reviewer.
- L 194 - “ITR sites” was changed to “ITR sites”.
- L 231 – “chromosome terminals” was changed to “chromosome termini”.
- Fig. 1b and 2b – the orientation of the bars was adjusted vertically as requested.
- References – corrections were made on both points indicated: n. 2 and n. 17.
Reviewer 2 Report
The work is well planned, done and written. From a cytogenetic point of view, it doesn't contain any particularly interesting results, but concerns a very important forage plant. Therefore, it may be interesting for breeders and researchers dealing with this species. In addition, the authors used a method still rarely used on plants (self-GISH), potentially very useful in research. I believe that work can be published after minor amendments.
M & M, 108-125: The authors write that “The hybridization solution was similar for FISH and GISH experiments with the exception of the concentration of probes”. Was salmon DNA used in both cases?
Results, 205-206: “Insets show the nucleolar chromosomes with signals of the 45S rDNA probe without being overlapped with DAPI”. In the Fig. 3 some signals seem to overlap.
Discussion, 263-266: My guess is that the authors intend to do this research. It would have been much better if the authors had already done it in this work, instead of dividing the research into small pieces.
References, 272: the lack of year.
Author Response
My response to the comments of Reviewer 2 is the following:
M & M: 108-125 – due to oversight I omitted that the salmon DNA was not used for GISH. Therefore I adjusted the sentence in L 114.
Results: 205-206 - I do not clearly understand the meaning of this comment, however I can confirm that the image from which I cut the nuclear chromosomes was not superimposed on the DAPI image to render more evident the green signals.
Discussion: 263-266 – about this comment I can say that I agree with Reviewer 2 but the time for realizing the complete study will be longer than planned. Therefore I prefer to publish the first results even though their value is less than that of the completed work.
References 272 – the year of publication has been added.
Reviewer 3 Report
The author present an interesting study, revealing for the first time, the presence and distribution of (TTTAGGG)n telomeric repeats on alfalfa chromosomes and the finding of the same repeats in the interstitial positions.
Author Response
My response to the comments of Reviewer 3 is the following:
Considering the overall comments of Reviewer 3 I would like to bring up some considerations.
The objective of this study was to provide a further knowledge of the chromosome structure of alfalfa (M. sativa subsp. sativa) rather than finding new chromosome markers. To realize my objective I applied a combination of in situ hybridization techniques.
The FISH mapping of telomeric repeats TTTAGGG demonstrated that telomeres of alfalfa are constituted by these sequences. Even though this feature has been already mentioned by Yu et al.(2013), as I reported in the Discussion, I would like to point out that the analysis of mitotic metaphase chromosomes hybridized by the telomeric probe (one of which is shown in Fig. 3) which I have carried out in this study has obtained more information on telomeres of alfalfa (in addition to confirming the presence of the TTTAGGG sequences), for example about the size of the sites and the intensity of the signals. In addition these features are clearly evident in the image in Fig. 3. As far as I know this is the first image of alfalfa telomeres so far published.
During the last years an increasing number of plant (and animal) species with interstitial telomeric repeats (ITRs) has been discovered. The increase of studies improved our knowledge on this topic leading to some hypotheses on the significance of these sequences. By means of the combined use of GISH and FISH techniques I could localize with precision the interstitial sequences and formulate hypotheses on the significance of their presence of this species.
Response to specific points.
Pag. 2 line 87 – I must clarify that the probe used was the plasmid pUc18 containing the Arabidopsis-like repeats TTTAGGG and that it was named pLt11 in the lab during the experiments. In M. and M. this point has been corrected.
Pag. 3 line 96 - this point, deriving from an oversight, has been corrected by deleting “several hours”.
Reviewer 4 Report
Review comments:
The manuscript “insight into the chromosome structure of the cultivated tetraploid alfalfa (Medicago sativa subsp. sativa L.) reports the efforts of Egizia Falistocco to distinguish specific chromosomal segments of alfalfa using FISH/GISH techniques. As expecting, GISH using total genomic DNA from alfalfa (self-GISH) produce a pattern of in situ signals similar to those obtained by C-banding techniques. FISH using a telomeric probe from Arabidopsis thaliana (pLt11) showed clear signals on the end of all chromosomes arms. In addition, novel interstitial signals were observed on numerous chromosomes.
The manuscript provides nice figures of alfalfa chromosomes. However, the self-GISH pattern, similar to the C-banding pattern previously reported by the author (Falistocco et al., 1995), doesn´t provide a new chromosomal marker in alfalfa. The presence of the TTTAGGG sequence typical of plants telomeres has been previously reported on alfalfa chromosomes (Yu et al., 2013). The only new data present in this study regards the presence of interstitials clusters of telomeric repeats in alfalfa chromosomes. However, the presence of interstitials telomeric repeats it is no rare in many plants and animals species. Thus, the main criticism is that there is not so much novelty reported.
Specific points:
Materials and Methods.
- Pg 2, line 87. The plasmid pLt11 contained the telomeric repeats… however, any reference to pLt11 is missing
- Pg 3. Line 96. ….fixed overnight in 3:1 absolute…..but then in line 97 author state ….at room temperature for several hours. How were roots fixed overnight or several hours?
Author Response
My response to the comments of Reviewer 4 is the following:
Considering the overall comments of Reviewer 3 I would like to bring up some considerations.
The objective of this study was to provide a further knowledge of the chromosome structure of alfalfa (M. sativa subsp. sativa) rather than finding new chromosome markers. To realize my objective I applied a combination of in situ hybridization techniques.
The FISH mapping of telomeric repeats TTTAGGG demonstrated that telomeres of alfalfa are constituted by these sequences. Even though this feature has been already mentioned by Yu et al.(2013), as I reported in the Discussion, I would like to point out that the analysis of mitotic metaphase chromosomes hybridized by the telomeric probe (one of which is shown in Fig. 3) which I have carried out in this study has obtained more information on telomeres of alfalfa (in addition to confirming the presence of the TTTAGGG sequences), for example about the size of the sites and the intensity of the signals. In addition these features are clearly evident in the image in Fig. 3. As far as I know this is the first image of alfalfa telomeres so far published.
During the last years an increasing number of plant (and animal) species with interstitial telomeric repeats (ITRs) has been discovered. The increase of studies improved our knowledge on this topic leading to some hypotheses on the significance of these sequences. By means of the combined use of GISH and FISH techniques I could localize with precision the interstitial sequences and formulate hypotheses on the significance of their presence of this species.
Response to specific points.
Pag. 2 line 87 – I must clarify that the probe used was the plasmid pUc18 containing the Arabidopsis-like repeats TTTAGGG and that it was named pLt11 in the lab during the experiments. In M. and M. this point has been corrected.
Pag. 3 line 96 - this point, deriving from an oversight, has been corrected by deleting “several hours”.